# Molecular Epidemiology of Multidrug-Resistant *Pseudomonas aeruginosa* Acquired in a Spanish Intensive Care Unit: Using Diverse Typing Methods to Identify Clonal Types

**DOI:** 10.3390/microorganisms10091791

**Published:** 2022-09-06

**Authors:** Marta Adelantado Lacasa, Maria Eugenia Portillo, Joaquin Lobo Palanco, Judith Chamorro, Carmen Ezpeleta Baquedano

**Affiliations:** 1Microbiology Area, Laboratory Department, Hospital Reina Sofía, 31500 Tudela, Spain; 2Instituto de Investigación Sanitaria de Navarra—IdiSNA, 31008 Pamplona, Spain; 3Clinical Microbiology Department, Hospital Universitario de Navarra, 31008 Pamplona, Spain; 4Intensive Care Unit, Hospital Universitario de Navarra, 31008 Pamplona, Spain; 5Preventive Medicine Department, Hospital Universitario de Navarra, 31008 Pamplona, Spain

**Keywords:** genotyping, high-risk clone, molecular characterization, multidrug-resistant, ST175, *Pseudomonas aeruginosa*, whole-genome sequencing

## Abstract

The increasing number of infections from multidrug-resistant *P. aeruginosa* (MDRPA) has compromised the selection of appropriate treatment in critically ill patients. Recent investigations have shown the existence of MDRPA global clones that have been disseminated in hospitals worldwide. We aimed to describe the molecular epidemiology and genetic diversity of the MDRPA acquired by Intensive Care Units (ICU) patients in our hospital. We used phenotypic methods to define the MDRPA and molecular methods were used to illustrate the presence of carbapenemase encoding genes. To characterize the MDRPA isolates, we used MALDI-TOF biomarker peaks, O-antigen serotyping, and multi-locus sequence typing analyses. Our data show that the most widely distributed MDRPA clone in our ICU unit was the ST175 strain. These isolates were further investigated by the whole-genome sequencing technique to determine the resistome profile and phylogenetic relationships, which showed, as previously described, that the MDR profile was due to the intrinsic resistance mechanisms and not the carbapenemase encoding genes. In addition, this study suggests that the combination of environmental focus and cross-transmission are responsible for the spread of MDRPA clones within our ICU unit. Serotyping and MALDI-TOF analyses are useful tools for the early detection of the most prevalent MDRPA clones in our hospital. Using these methods, semi-directed treatments can be introduced at earlier stages and healthcare professionals can actively search for environmental foci as possible sources of outbreaks.

## 1. Introduction

*Pseudomonas aeruginosa* is a frequent source of hospital-acquired infections. These non-fermenting, gram-negative bacteria can cause severe infections in immunocompromised patients (especially in neutropenic patients) and patients in Intensive Care Units (ICU) that may be associated with high rates of mortality. Given hospital environments, especially those with high rates of humidity, can act as a reservoir for *P. aeruginosa* [1,2]. These environmental sources may be the foci for the dissemination of this pathogen in common-source outbreaks.

Increasing amounts of infections from multidrug-resistant *P. aeruginosa* (MDRPA) has compromised the selection of appropriate treatments in critically ill patients, increasing morbidity and mortality in these patients. *P. aeruginosa* has a non-clonal population structure with a small number of widespread selected clones [3]. These clones have the capacity to spread among hospitals and travel within hospital units, and they are responsible for intra-hospital outbreaks worldwide. These clones are referred to as high-risk clones (HRCs). There is a strong association between the HRCs and the defined multidrug resistance profiles [4]. The detection of HRCs in clinical microbiology laboratories is an important task for infection control and appropriate treatment guidance.

A variety of molecular methods have been used to type *P. aeruginosa* strains and to identify these HRCs. Pulse field gel electrophoresis (PFGE) is a method that represents the “gold standard”; it is a method with great technical complexity that requires a specialized technical staff. In the last few years, highly discriminatory and replicable molecular typing methods have been developed as a method for multi-locus sequence typing (MLST) [5]. However, MLST is not practical to implement for guiding antipseudomonal therapy in microbiological diagnostic laboratories because it is time consuming. Even though the PFGE method is useful to investigate MDRPA outbreaks, MLST is the method that makes it possible to investigate the strain relatedness over long periods of time, and it is the most widely accepted “gold standard” technique for the definition of MDRPA epidemic clones.

Recently, a method using MALDI-TOF has been developed to identify HRCs in an accurate and quick manner [6]. This method is able to identify the ST175 HRC with a high sensitivity and specificity by looking for two MALDI-TOF biomarker peaks [6,7]. ST175 is a frequent strain in Europe, particularly in Spain and France [3,4,8,9]. Furthermore, it has a defined multi-resistance profile that is typically only susceptible to ceftolozane/tazobactam, ceftazidime/avibactam, amikacin, and colistin. In addition to the association with these two peaks, the ST175 strain is associated with serotype O4 [7]. O-antigen serotyping is a rapid and simple procedure that, despite not being as discriminatory as the MLST and PFGE methods, still might be useful for the presumptive detection of some HRCs, as O-antigens have been associated with certain STs and multidrug-resistant profiles [10].

Over the recent years, whole-genome sequencing (WGS) has become an increasingly prevalent and cost-effective tool in microbiology laboratories. With new sequencing technologies and new bioinformatics analysis platforms, it has become feasible to perform WGS techniques and interpret their results with the help of less specialized personnel than had been previously required. WGS techniques provide great advantages in the molecular characterization of MDRPA by providing information about the resistome, molecular typing, and population structure.

We aimed to describe the strain characterization of MDRPA that was acquired by ICU patients in our hospital by using different methods to develop specific control strategies and to help guide early antimicrobial therapy. Moreover, we used WGS techniques to perform an epidemiological analysis of a cohort of ST175 MDRPA that was isolated in our ICU unit. Additionally, we reported an MDRPA-presumed outbreak during this study, which we aimed to further investigate.

## 2. Materials and Methods

### 2.1. Study Design

The study was performed from January to December 2019. Patients that were admitted into the ICU in the Hospital Universitario de Navarra (HUN) during this period had perianal culture samples obtained at the time of ICU admission and once per week during their hospitalization. The samples were obtained as part of a Zero Resistance surveillance program, which attempts to look for multidrug-resistant microorganism carriers.

We classified two patient types: (a) admission-positive patients that had positive cultures at the time of ICU admission for MDRPA; and (b) acquisition-positive patients that had negative cultures at the time of ICU admission but a subsequent positive culture for MDRPA.

In response to an MDRPA outbreak in the ICU, we initiated a study in which swab samples from the ICU tap drains were additionally collected for the detection of MDRPA and carbapenemase-producing isolates.

### 2.2. Microbiological Methods

Perianal surveillance swabs and tap drain samples were grown on selected chromogenic media (CHROMID^®^CARBA SMART, bioMérieux, Marcy-l’Étoile, France; and CHROMID^®^ ESBL, bioMérieux, Marcy-l’Étoile, France). Isolates suggestive of *P. aeruginosa* were identified with a MALDI-TOF analysis (MALDI Biotyper, Bruker Daltonics, Bremen, Germany). Antimicrobial susceptibility tests were performed by disk diffusion tests according to the European Committee on Antimicrobial Susceptibility Testing (EUCAST) guidelines (“Breakpoint tables for interpretation of MICs and zone diameters”. Version 9.0, 2019. http://www.eucast.org (accessed on 3 January 2020)). *P. aeruginosa* isolates were classified into non-resistant and multidrug-resistant (MDR) according to their antimicrobial susceptibility pattern and according to previously described criteria [11].

### 2.3. Carbapenemase Encoding Genes Identification

The MDRPA carbapenem-resistant isolates were further investigated by molecular methods to determine the presence of carbapenemase encoding genes. We performed in-house multiplex PCR tests for the detection of commonly identified carbapenemases in our media (KPC, IMP, VIM, NDM, and OXA-48) (primer sequences are shown in Table 1, annealing temperature of 61.5 °C) and for the detection of the less common class A (BIC, GES, IMI, NMCA, and SME) (primer sequences are shown in Table 2, annealing temperature of 54 °C) and class B carbapenemases (AIM, SIM, DIM, GIM, and SPM) (primer sequences are shown in Table 3, annealing temperature of 60.5 °C).

### 2.4. Strain Characterization

We characterized the MDRPA isolates by two typing methods:-O-type antigen serotyping by the process of agglutination with O1, O4, O11, and O12 antisera (Bio-Rad Laboratories, Redmond, WA, USA)-The MLST method, as described previously in [5]

In all of the MDRPA isolates, we investigated the presence of 6911 and 7359 m/z peaks in the MALDI-TOF spectra. These two biomarker peaks can detect early ST175 HRCs as previously described in Cabrolier et al. (2015) and Mulet et al. (2019) [6,7].

### 2.5. Whole-Genome Sequencing

The MDRPA strain were identified as an ST175 clone and were analysed by the WGS technique, performed using the NEBNext Ultra II FS DNA Library Prep Kit (New England Biolabs Inc., Ipswich, MA, USA) in an iSeq 100 instrument (Illumina Inc., San Diego, CA, USA). The sequencing data were analysed with the easy-to-use, fully integrated web-based software application EPISEQ^®^ CS, Version 1 (bioMérieux, Marcy-l’Étoile, France). The application is based on reference-free approach with automated workflow (https://www.biomerieux-episeq.com/cs-how-it-works). In brief terms, it checks the quality of sequencing data and uses the open-source algorithm SPAdes for genome assembly [12]. Species identity, initially selected by the customer, and potential intra- and inter-species contamination are checked. Genomic strain characterization is performed through the generation of MLST results and prediction of antimicrobial resistance genes (resistome). Allele calling is performed in a proprietary whole-genome MLST (wgMLST) scheme (15,143 total loci, including 1480 core loci, for *P. aeruginosa*). We performed a wgMLST-based epidemiological analysis of the sequenced ST175 MDRPA using EPISEQ^®^ CS application [13]. We constructed a minimum spanning tree based on the wgMLST allele differences between the isolates. For clustering, samples with no differences were grouped into the same node, and the maximum distance between nodes was customized to >14 allele differences. Furthermore, we drew up a dendrogram which reflected the relationships between the isolates. The EPISEQ^®^ CS application constructs the matrix using both a similarity coefficient (match or no match between the wgMLST allele numbers) between the samples and a clustering algorithm (the unweighted pair-group method using arithmetic averages, or UPGMA, algorithm) [13]. We established the high threshold similarity value at 98% and the low threshold value at 95%, and we calculated the number of clusters based on these cut-off values. Isolates with a similarity score above 98% were considered as probably related; isolates with a similarity score between 95 and 98% were defined as possibly related; and isolates with a similarity score under 95% were considered as probably unrelated.

### 2.6. Outbreak Study

The MDRPA isolated from perianal surveillance swabs and ICU tap drains that were suggestive of belonging to an outbreak and were epidemiologically related were sent to the National Microbiology Center (Majadahonda, Spain) for further investigation by the PFGE and MLST techniques. All isolates were grown in Mueller–Hinton agar overnight. Once the PFGE-agarose blocks were performed, they were digested with BcuI, and the resulting fragments were separated by using electrophoresis in 1% agarose gels with the CHEF-DR II device (Bio-Rad Laboratories, Hercules, CA, USA) for 21 h, with switch times ranging from 5 s to 40 s in a Tris-borate ethylenediaminetetraacetic acid (TBE) buffer. Photographic images of the gels were digitally saved with the Geldoc EQ system (Bio-Rad Laboratories, Hercules, CA, USA). The DNA macrorestriction patterns were compared in order to determine band similarity, and they were interpreted according to the criteria established by Tenover et al. [14] for defining pulsed-field type clusters. Two isolates were considered to be indistinguishable and genetically related (clones) when the restriction patterns had the same number of bands and those corresponding bands were of the same apparent size. Two isolates were considered to be closely related when the number of genetic differences was a single event and their restriction pattern differed in 2–3 fragments. Isolates that were indistinguishable or closely related were considered as part of (or probably part of) the outbreak.

We determined the acquisition route for patients who acquired an MDRPA strain based on the PFGE strain type and epidemiological relatedness.

A representative isolate of each cluster was subjected to further investigation by the MLST and molecular methods to determine the presence of carbapenemase encoding genes.

## 3. Results

### 3.1. Multidrug-Resistant P. aeruginosa at the ICU Unit: Patients’ Admission vs. Acquisition Classification

Our cohort consisted of 1867 swabs with 255 isolated cases of *P. aeruginosa,* of which 64 (25%) were MDRPA. From these MDRPAs, we randomly chose 30 isolates from 28 patients and studied them in further detail. The patients were categorized as follows: 3 patients (10.7%) that had positive cultures at the time of ICU admission and were classified as admission-positive patients; and 25 patients (89.3%) that we had obtained negative perianal cultures for MDRPA at the time of ICU admission, but a subsequent weekly culture that was positive for MDRPA; such patients were defined as acquisition-positive patients. The average number of days for the acquisition of MDRPA in our unit was 26.5 ± 5.55 days (a range of 7–78 days).

### 3.2. MDRPA Acquisition-Positive Patients’ Strain Characterization

In acquisition-positive patients, 27 MDRPA strains were isolated and characterized. Fifteen (55.6%) isolates were identified to be ST175 on the MLST test. All of them were of serogroup O4, had 6911 and 7359 m/z MALDI-TOF biomarkers peaks, and were not carbapenemase producers, despite their carbapenem-resistant phenotype. Two (7.4%) isolates were identified to be CC235 and showed positive agglutination with the O11 antisera. These isolates were carbapenem-resistant, but non-carbapenemase producers. Three (11.1%) isolates were identified to be ST253; they were VIM-2 carbapenemase producers and did not show agglutination with the O1, O4, O11, or O12 antisera. Seven (25.9%) isolates were genetically heterogeneous non-carbapenemase producers and did not agglutinate with the antisera (Figure 1).

### 3.3. ST175 Clone MDRPA Whole-Genome Sequencing

Eighth out of the fifteen ST175 MDRPA strains were further analysed by the WGS technique. We performed the MLST technique, resistome profiling, and epidemiological analysis. The WGS analysis confirmed MLST for ST175 in five isolates. The minimum spanning tree based on the allelic differences (Figure 2) showed two isolates forming a cluster (Cluster 1, 11 allele differences). One isolate formed Cluster 2 and was closely related to Cluster 1 (21 allele differences). Five isolates formed five different clusters. Cluster 3 and 4 were moderately distant (ranging 90–100 allele differences) from Cluster 2, and Cluster 5 was distant from Cluster 1. Cluster 6 and 7 were more diverse compared with the other clusters (>4000 allele differences).

The dendrogram reflected one cluster comprising four isolates that were probably related (Cluster 1 dendrogram), two isolates that were possibly related, and two isolates that were probably not related (Figure 3).

The resistome analysis showed that isolates that were probably related presented the same resistome (Table 4). No horizontally-transmitted genes associated with carbapenemase production were detected. Resistance to carbapenems could be conferred by the following detected alleles in the four isolates that were probably related and in the two that were possibly related: the blaOXA-50 and blaPDC families. Resistance to carbapenems in the two isolates that were probably not related was conferred by the blaGES genes and other blaOXA and blaPDC alleles, compared with the other six isolates (Table 4). Table 5 shows the mechanism of action and target drugs of each protein, deduced from the main resistance genes detected.

### 3.4. Outbreak Study

In the context of a suspected ICU MDRPA outbreak, we sent 12 strains to the National Microbiology Center for PFGE testing: 8 strains were isolated from 7 ICU patients and 4 strains were isolated from ICU tap drains. The PFGE band patterns showed an epidemiologically complex situation. Cluster 1 comprised a predominant/major clone compound by four strains that were isolated from four patients and were non-carbapenemase producers, the ST175 strain and O4 serogroup. Cluster 2 was a minor-clone, and comprised VIM-2 carbapenemase ST253 strains isolated from two patients (two strains) and from one tap drain (one strain). Another minor-clone formed Cluster 3, and was formed by VIM-2 carbapenemase ST253 strains that were isolated from one patient (one strain) and from three tap drains (three strains). One MDRPA strain from a patient was a non-carbapenemase producer and genetically heterogeneous, therefore not genetically related and unable to be classified inside a PFGE cluster.

## 4. Discussion

This study showed the strain characterization of MDRPA acquired by patients admitted to the ICU unit. Nearly 55% of the ICU MDRPA acquired were of the ST175 strain and showed positive agglutination with the O4 antigen. Our data are in agreement with a national Spanish multicentre study [10], which showed that the most frequent clone among Spanish multidrug-resistant (MDR) isolates was the ST175 strain, the most frequent serotype was O4, and that the serotype was linked to widespread ST175 HRCs. The model described by Mulet et al. [7] proved that the combination of MALDI-TOF and O4 agglutination analyses could presumptively identify ST175 HRCs with high sensitivity and specificity. According to this model, our data show that the O4 serotype and 6911 and 7359 m/z MALDI-TOF biomarker peaks were present among all of the ST175 isolates, and that they were not found among the non-ST175 isolates. The early detection of the O4 serotype and the presumptive identification of the ST175 strain using MALDI-TOF testing may be useful for the early guiding of semi-empirical antibiotic treatment for *P. aeruginosa* in critically ill patients. None of the ST175 MDRPA isolates in this study were carbapenemase producers despite their carbapenem-resistant phenotype. No genes associated with carbapenemase production were detected by the in-house PCR tests. This finding was validated by the WGS analysis. The multidrug-resistant phenotype associated with the ST175 strain is mainly mutational [3,8], represented by a combination of specific AmpR, OprD inactivation, and quinolone resistance-determining regions (QRDR) mutations. The WGS analysis confirmed this finding, as the MDRPA ST175 resistome showed that there is a combination of genes responsible for the multi-resistant phenotype (mainly OXA-50, PDC, crpP, and PmpM). The WGS analysis also confirmed that in our ICU unit, there is a homogeneous population of MDRPA ST175 isolates that are possibly related to each other and have the same origin.

According to our data, 11.1% of MDRPA isolates belonged to the ST253 strain and were VIM-2 carbapenemase producers. The ST253 strain has a worldwide distribution, but it is not particularly associated with any resistance mechanism and it is therefore not considered one of the main HRCs. However, it has been acquiring resistance determinants, including metallo-beta-lactamases (MBL) in some hospitals; for instance, in Catalonia it is indeed associated with VIM-1 carbapenemase [8,15]. Our results agree with these past findings. In conclusion, the ST253 strain is a prevalent clone that is beginning to be associated with MDR profiles. In addition, it is a rather virulent clone, testing positive for exoU, a well-known virulence gene.

The ST235 strain is the most widely distributed HRC worldwide [16]. The ST235 strain is the founder clone of the CC235 clonal complex, and it is associated with the O11 antigen [3]. Association of the ST235 strain with transferrable resistance genes, particularly with Class B carbapenemases, is a matter of concern. VIM-2 is one of the most geographically widespread beta-lactamases among the ST235 isolates [3]. In our study, only a few (7.4%) isolates were of the CC235 clonal complex, with ST175 being the dominant clone. All of the CC235 isolates showed positive agglutination with the O11 antisera, but none were carbapenemase producers despite their carbapenem-resistant phenotype, in contrast to the evidence.

Within this cohort, over 80% of the MDRPA isolated from patients admitted to our ICU unit was acquired intra-ICU. A previous study showed that almost 50% of the imipenem-resistant *P. aeruginosa* were acquired during ICU hospitalization [17]. Although our patients temporarily coincided in the ICU, it is not possible to discern whether it was patient-to-patient transmission or whether there was an active environmental source. However, the results of our outbreak study suggest that there is the possibility of an environmental source. In fact, for three (14%) patients the VIM transmission was through tap drains. Several studies have demonstrated that hospital water environments can act as a reservoir for multidrug-resistant organisms, causing hospital-acquired infections [18,19,20,21,22]. Encouraged by these results, we implemented a series of measures based on the surveillance and disinfection of environmental sources, especially in areas with humid atmospheres. For appropriate long-term clearance, we replaced colonized water reservoirs. In addition, we reviewed patient hygiene routines and medication preparation procedures and reinforced infection control measures. Understanding the epidemiology and transmission modes of the MDRPA is necessary to implement measures that control the transmission.

In conclusion, due to the high prevalence of the ST175 MDRPA isolates in our local media, it is important to actively monitor for this HRC. O-antigen serotyping and MALDI-TOF are useful methods for the early detection of this clone. Using these techniques, semi-directed treatment can be introduced early. WGS analyses are becoming increasingly important in Microbiology Laboratories. Their implementation and the interpretation of their results using new, web-based informatics pipelines have been made progressively feasible.

## Figures and Tables

**Figure 1 microorganisms-10-01791-f001:**
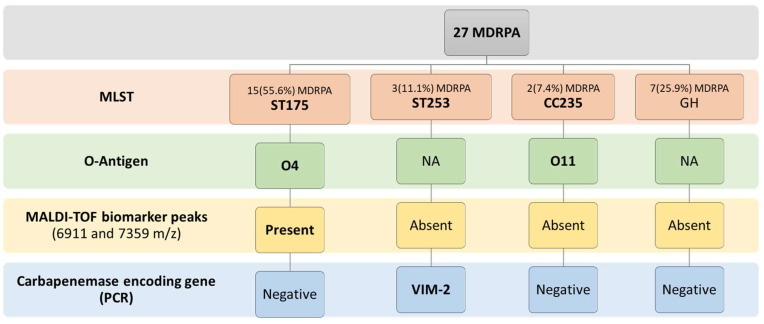
The MLST, O-antigen determination, MALDI-TOF biomarker peaks, and PCR carbapenemase encoding genes results of the MDRPA strains that were isolated from acquisition-positive ICU patients. MDRPA: multidrug-resistant *P. aeruginosa*; GE: genetically heterogeneous; NA: no agglutination.

**Figure 2 microorganisms-10-01791-f002:**
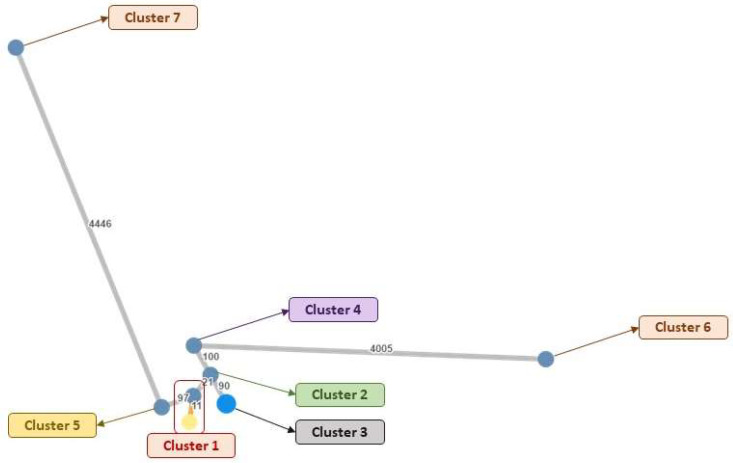
A minimum spanning tree based on the number of allelic differences between the isolates. The number of allelic differences are indicated on the lines connecting the isolates.

**Figure 3 microorganisms-10-01791-f003:**
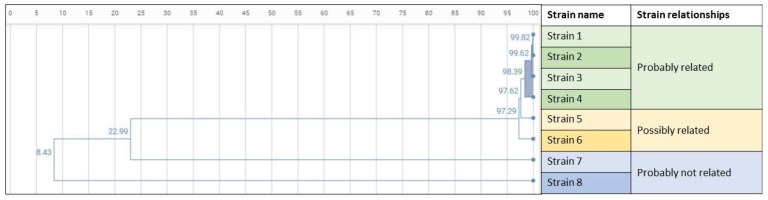
Dendrogram. High and low similarity coefficient cut-offs were established at >98% and 95%, respectively. Four isolates were classified as probably related and formed Cluster 1. Two isolates were considered possibly related and another two isolates were considered probably not related.

**Table 1 microorganisms-10-01791-t001:** The primers used for the amplification of commonly identified carbapenemases in our media (KPC, IMP, VIM, NDM, and OXA-48).

Gene	Primer	Primer Sequence	Amplified Fragment Size (pb)	Anneal. T (°C)
KPC	Forward: KPCseq-F	5′-TGTCACTGTATCGCCGTC-3′	881	61.5
	Reverse: KPCseq-R	5′-TTACTGCCCGTTGACGCC-3′		
IMP	Forward: IMP-up	5′-GAAGGCGTTTATGTTCATAC-3′	587	
	Reverse: IMP-dn	5′-GTAAGTTTCAAGAGTGATGC-3′		
VIM	Forward: VIM-1	5′-GTTTGGTCGCATATCGCAAC-3′	389	
	Reverse: VIM-2	5′-AATGCGCAGCACCAGGATAG-3′		
NDM	Forward: NDMseq-F	5′-CCATGCGGGCCGTATGAGTGATTG-3′	768	
	Reverse: NDMseq-R	5′-TCGCGAAGCTGAGCACCGCATTAG-3′		
OXA-48	Forward: OXA48seq-F	5′-TGCGTGTATTAGCCTTATCG-3′	785	
	Reverse: OXA48seq-R	5′-TTTTTCCTGTTTGAGCACTTC-3′		

**Table 2 microorganisms-10-01791-t002:** The primers used for the amplification of less common class A carbapenemases (BIC, GES, IMI, NMCA, SME).

Gene	Primer	Primer Sequence	Amplified Fragment Size (pb)	Anneal. T (°C)
BIC	Forward: BIC-F	5′-TATGCAGCTCCTTTAAGGGC-3′	537	54
	Reverse: BIC-R	5′-TCATTGGCGGTGCCGTACAC-3′		
GES	Forward: GES1-A	5′-ATGCGCTTCATTCACGCAC-3′	863	
	Reverse: GES1-B	5′-CTATTTGTCCGTGCTCAGG-3′		
IMI	Forward: IMI-up	5′-GTCACTTAATGTAAAACC-3′	873	
	Reverse: IMI-dn	5′-TTAAGGTTATCAATTGCG-3′		
NMCA	Forward: NMCA-up	5′-GTCACTTAATGTAAAGCA-3′	869	
	Reverse: NMCA-dn	5′-GGTTATCAATTGCAATTC-3′		
SME	Forward: SME-up	5′-CGGCTTCATTTTTGTTTA-3′	954	
	Reverse: SME-dn	5′-CAATTGCCTGAATTGCAAT-3′		

**Table 3 microorganisms-10-01791-t003:** The primers used for the amplification of less common B carbapenemases (AIM, SIM, DIM, GIM, SPM).

Gene	Primer	Primer Sequence	Amplified Fragment Size (pb)	Anneal. T (°C)
AIM	Forward: AIM-F	5′-CTGAAGGTGTACGGAAACAC-3′	322	60.5
	Reverse: AIM-R	5′-GTTCGGCCACCTCGAATTG-3′		
SIM	Forward: SIM-F	5′-TACAAGGGATTCGGCATCG-3′	570	
	Reverse: SIM-R	5′-TAATGGCCTGTTCCCATGTG-3′		
DIM	Forward: DIM-F	5′-GCTTGTCTTCGCTTGCTAACG-3′	699	
	Reverse: DIM-3	5′-CGTTCGGCTGGATTGATTT-3′		
GIM	Forward: GIM-up	5′-ACTTGTAGCGTTGCCAGC-3′	722	
	Reverse: GIM-dn	5′-AATCAGCCGACGCTTCAG-3′		
SPM	Forward: SPM-1A	5′-CTGCTTGGATTCATGGGCGC-3′	784	
	Reverse: SPM-1B	5′-CCTTTTCCGCGACCTTGATC-3′		

**Table 4 microorganisms-10-01791-t004:** The deduced proteins from the resistance genes detected by the WGS analysis in the ST175 MDRPA isolates. The variants that were detected are specified.

TARGET DRUGS	Probably Related	Possibly Related		Probably Not Related	
	Strains 1–4	Strain 5	Strain 6	Strain 7	Strain 8
Cephalosporins, Carbapenems	OXA-50	OXA-50	OXA-50	OXA-396 (OXA-50 family)	OXA-846 (OXA-50 family)
	PDC-221 PDC-222 PDC-226 PDC-321		PDC-221	PDC-113 PDC-141 PDC-157 PDC-203 PDC-254 PDC-261 PDC-307 PDC-338 PDC-40	PDC-11
					GES-1 GES-11 GES-26
Aminoglycosides, Quinolones	PmpM	PmpM	PmpM	PmpM	PmpM
Quinolones	crpP	crpP	crpP	crpP	crpP

PDC: Pseudomonas-derived cephalosporinase; GES: Guiana extended-spectrum; crpP: ciprofloxacin resistance protein.

**Table 5 microorganisms-10-01791-t005:** Target drugs and the mechanism of action of the proteins, deduced from the main resistance genes detected.

Deduced Proteins from Resistance Genes	Classification	Target Drugs
OXA-50 family (OXA-50, OXA-396, OXA-846)	Class D beta-lactamase	Cephalosporins Piperacillin-tazobactam Meropenem
PDC	Class C beta-lactamase	Cephalosporins
GES	Class A beta-lactamase	Cephalosporins Carbapenems
crpP	Phosphorylase	Quinolones
PmpM	Multidrug efflux pump	Aminoglycosides Quinolones

## Data Availability

Data sharing not applicable.

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
