# Peer review of "Molecular Epidemiology of Multidrug-Resistant Pseudomonas aeruginosa Acquired in a Spanish Intensive Care Unit: Using Diverse Typing Methods to Identify Clonal Types"

_microorganisms, 2022, doi:10.3390/microorganisms10091791_

Round 1

Reviewer 1 Report

The aim of the study was to investigate molecular epidemiology of multidrug resistant Pseudomonas aeruginosa acquired in a Spanish Intensive Care Unit. Thus, the results might be of local interest. The Authors applied diverse typing methods to identify clonal types and reached their goal but the methods or results are not described sufficiently. Moreover, they investigated the presence of the most common antimicrobial resistance mechanisms genes amongst  Pseudomonas aeruginosa rods. But again, the methodology description and graphic presentation of the results are absent.

My concerns are as follows:

·       Keywords should be listed in alphabetical order, in my opinion.

·       Introduction should be re-written and definitely divided into few separate sections.

·       What is IPM carbapenemase?

·       Lack of description of the applied Methods, e.g. PCR.

·       Lack of any figures showing the results of PCR etc.

·       How did you calculate 80% similarity of the strains using Tenover et al. criteria?

·       What does 98% similarity mean?

·       The dendrogram figure is repeated.

·       “Swap” needs correction?

·       Figure 3 is illegible.

·       Results – it would be useful to add the time necessary for P. aeruginosa strains acquisition during hospitalization.

·       It should be explained/discussed why O-antigen could not be determined.

·       Discussion – first sentence, is it precise that  This study analyses the molecular epidemiology of MDRPA acquired by patients admitted to ICU unit”?

·       Lack of any statistical analysis.

·       The whole methodology is presented very superficially; moreover there is a lack of any graphics for PCR results showing presence of the genes linked to antimicrobial resistance.

·       The Discussion section is quite superficial.

The study may present a really interesting observations but it should be profoundly re-written to show and underline or indicate the most important aspects of the research.

Author Response

Dear Reviewer,

First of all, we would like to express our gratitude for review our manuscript. We really appreciate your comments and suggestions of change. We have checked your reviewer report form and we have done the suggested modifications. We consider that the edits we have introduced have served to improve the scientific quality of the original manuscript. The authors would like to response your appreciate comments.

The aim of the study was to investigate molecular epidemiology of multidrug resistant Pseudomonas aeruginosa acquired in a Spanish Intensive Care Unit. Thus, the results might be of local interest. The Authors applied diverse typing methods to identify clonal types and reached their goal but the methods or results are not described sufficiently. Moreover, they investigated the presence of the most common antimicrobial resistance mechanisms genes amongst Pseudomonas aeruginosa rods. But again, the methodology description and graphic presentation of the results are absent.

Point 1: Keywords should be listed in alphabetical order, in my opinion.

Response 1: We have listed keywords in alphabetical order.

Point 2: Introduction should be re-written and definitely divided into few separate sections.

Response 2: We have separated Introduction in sections and we have made some modifications trying to improving it.

Point 3: What is IPM carbapenemase? “Swap” needs correction?

Response 3: We have idintified some spell and grammar erros and we have modified them.

Point 4: Lack of description of the applied Methods, e.g. PCR. Lack of any figures showing the results of PCR etc. The whole methodology is presented very superficially; moreover there is a lack of any graphics for PCR results showing presence of the genes linked to antimicrobial resistance.

Response 4: We have added tables containing the sequences of the primers used for the amplification of each gene. Annealing temperatures for each multiplex PCR are specified in the text now. Moreover, we have completed the description of the different methodology sections to clarify possible doubts. We have completed the section results with some changes. These changes affect to the presentation of some data. We have introduced one Figure (1) to explain better strain characterization and PCR results.

Point 5: How did you calculate 80% similarity of the strains using Tenover et al. criteria?

Response 5: We have detected an error in the explanation of the interpretation of PFGE results and have corrected it according to Tenover et al. criteria.

Point 6: What does 98% similarity mean?

 Response 6: We have completed the explanation of the rationale behind the EpiSeq pipeline. We have explained the basis of minnimun spanning tree and dendrogram construction to improve comprhension of the results.

Point 7: It should be explained/discussed why O-antigen could not be determined.

Response 7: We identified and expression error. We wanted to say that the agglutination with O1, O4, O11 and O12 antisera was not observed. We have changed it in the manuscript.

Point 8: The dendrogram figure is repeated. Figure 3 is illegible.

Response 8: We have made changes in the text and in Figures 2 and 3 and we have replaced an image by a table to better clarify the WGS and resistome results.

Point 9: Discussion – first sentence, is it precise that  “This study analyses the molecular epidemiology of MDRPA acquired by patients admitted to ICU unit”? The Discussion section is quite superficial.

Response 9: We have done some changes Discussion section to improve the scientific quality of our explanations.

Point 10: Results – it would be useful to add the time necessary for P. aeruginosa strains acquisition during hospitalization.

Response 10: We appreciate this suggestion and consider that it would be relevant to our research to know how long it takes for patients in our ICU to become colonised. We have calculated this data and added in Results section.

Point 11: Lack of any statistical analysis.

Response 11: Our study has a descriptive structure and it is not our aim to compare typing techniques. For this reason we did not initially consider necessary to add statistical analysis.

Finally, we added line numbers to the manuscript to facilitate the revision of the manuscript. We have performed an English grammar revision of the manuscript to improve English language and style.

Reviewer 2 Report

This manuscript “Molecular epidemiology of multidrug resistant Pseudomonas aeruginosa acquired in a Spanish intensive care unit: using diverse typing methods to identify clonal types” by Adelantado et al. studied the molecular epidemiology of MDRPA acquired by ICU patients under clinical settings. Resistome in analyzing antibiotic resistance genes across different strains are appreciated, and revisions clarifying the importance of the findings will be needed.

1. Method 2.2, please describe the PCR primer and conditions used.

2. For all reagents used in this study, please clarify their company and location.

3. I do not know why no line numbers are shown in the manuscript. It would help the review process.

4. Introduction line 2: gram-negative

5. Introduction line 21: grammar mistake.

6. Change figure x in the manuscript into Figure x.

Author Response

Dear Reviewer,

First of all, we would like to express our gratitude for review our manuscript. We really appreciate your comments and suggestions of change. We have checked your reviewer report form and we have done the suggested modifications. We consider that the edits we have introduced have served to improve the scientific quality of the original manuscript. The authors would like to response your appreciate comments.

This manuscript “Molecular epidemiology of multidrug resistant Pseudomonas aeruginosa acquired in a Spanish intensive care unit: using diverse typing methods to identify clonal types” by Adelantado et al. studied the molecular epidemiology of MDRPA acquired by ICU patients under clinical settings. Resistome in analyzing antibiotic resistance genes across different strains are appreciated, and revisions clarifying the importance of the findings will be needed.

Point 1: Method 2.2, please describe the PCR primer and conditions used.

Response 1: We have added tables containing the sequences of the primers used for the amplification of each gene. Annealing temperatures for each multiplex PCR are specified in the text now. Moreover, we have complete the description of the different methodology sections to clarify possible doubts.

Point 2: For all reagents used in this study, please clarify their company and location.

Response 2: We have specified the location of the company for all the reagents used.

Point 3: I do not know why no line numbers are shown in the manuscript. It would help the review process.

Response 3: The line numbers are now shown to facilitate the revision of the manuscript.

Point 4: Introduction line 2: gram-negative. Introduction line 21: grammar mistake. Change figure x in the manuscript into Figure x.

Response 4: We have performed an English grammar revision of the manuscript to improve English language and style.

We have completed the section results with some changes. These changes affect to the presentation of some data. We have introduced one Figure (1) to explain better strain characterization and PCR results, we have made changes in Figures 2 and 3 and we have replaced an image by a table to better clarify the resistome results.

Finally, we have done some changes in Introduction and Discussion sections to improve the scientific quality of our explanations.

Round 2

Reviewer 1 Report

N/A

Reviewer 2 Report

Thank you for responding to all my comments.
